**Data Availability Statement:** The study data cannot be shared publicly because the written consent obtained from the study participants did

# Associations between pre-stroke physical activity and physical quality of life three months after stroke in patients with mild disability

Melanie Zirnsak[1,2,3], Christine Meisinger[1,4], Jakob Linseisen[1,4], Michael Ertl[5], Philipp Zickler[5], Markus Naumann[5], Inge Kirchberger[1,2,4]*

**1** Chair of Epidemiology, University Augsburg, University Hospital Augsburg, Augsburg, Germany, **2** Institute for Medical Information Processing, Biometry and Epidemiology-IBE, Ludwig-Maximilians-Universität München, Munich, Germany, **3** Pettenkofer School of Public Health, Munich, Germany, **4** Independent Research Group Clinical Epidemiology, Helmholtz Zentrum München, German Research Center for Environmental Health, Neuherberg, Germany, **5** Department of Neurology and Clinical Neurophysiology, University Hospital Augsburg, Augsburg, Germany

* Inge.Kirchberger@med.uni-augsburg.de

## Abstract

### Background

Much is known about the association between physical activity and the occurrence of stroke. However, the evidence about the correlation between pre-stroke physical activity and post-stroke quality of life remains inconsistent. Thus, there is a high public health relevance to the topic.

### Aim

The aim of this study was to investigate the association between pre-stroke physical activity and physical quality of life after three months.

### Methods

Data arises from 858 patients with stroke included a prospective single-centre observational cohort study in Augsburg, Germany, between September 2018 and November 2019. The participants were recruited at the Department of Neurology and Clinical Neurophysiology, University Hospital of Augsburg after their stroke event. The level of physical activity was determined following the short form of the International Physical Activity Questionnaire at baseline. Physical quality of life was assessed three months after hospital discharge using the German version of the Stroke Impact Scale (SIS). A multiple linear regression model and a quantile regression were carried out.

### Results

A total of 497 patients were included in the analysis (mean age 69.6, 58.8% male), 26.2% had a high, 18.9% a moderate and 54.9% a low level of pre-stroke physical activity. Patients

not include a public availability of their data. Data are available from the data owner Chair of Epidemiology at the University of Augsburg (Website: http://www.uni-augsburg.de/med/epidemiologie, e-mail: epi-studiensekretariat@med.uni-augsburg.de) on request.

**Funding:** The author(s) received no specific funding for this work.

**Competing interests:** The authors have declared that no competing interests exist.

with high pre-stroke physical activity had a significantly better physical quality of life three months after stroke in the SIS physical domain (beta = 4.1) and in the SIS subdomains hand function (beta = 5.6), mobility (beta = 4.1) and activities of daily living (beta = 3.7). In the physical domain and the subdomain mobility, the effect was especially strong for persons with low physical quality of life after three months.

## Conclusion

Pre-stroke physical activity seems to have an important and positive association with physical quality of life after three months in patients with mild disability. Further studies are needed to confirm these results.

## Introduction

Stroke is one of the most common causes of disability in adults [1–3] and it is the second most frequent cause of death in Germany as well as on global scale [1, 2, 4]. Lifestyle factors like obesity, poor diet or physical inactivity are seen as major modifiable risk factors for stroke [5]. Regular physical activity decreases stroke incidence [6, 7] and was associated with better cognitive function [8] and even fewer symptoms of depression [9] in those affected. There are a few studies which reported a significant association between pre-stroke physical activity and post-stroke functional status as assessed by the National Institutes of Health Stroke Scale (NIHSS), the Modified Ranking Scale (mRS) and the Barthel Index [10–13]. In addition, low level of physical activity before stroke predicted low physical activity after stroke [14]. This is important, since higher levels of post-stroke physical activity are related with better physical function as well as better quality of life [15]. However, little is known about the correlation between pre-stroke physical activity and post-stroke outcomes such as health-related quality of life (HRQOL), which is considered to be considerably lower in stroke survivors than population norm [16]. Since physical activity is an essential target of stroke rehabilitation, further knowledge about the relation of pre-stroke physical activity and post-stroke HRQOL could be used to identify patients at risk for inactivity and impaired HRQOL after stroke.

Therefore, this study aims to investigate the associations between pre-stroke physical activity and physical quality of life after three months using longitudinal data from a prospective single-centre observational cohort study in Augsburg, Germany. In detail, two objectives were met: (1) to analyse the associations between pre-stroke physical activity and the Stroke Impact Scale (SIS) physical domain after three months and (2) to analyse the associations between pre-stroke physical activity and the SIS subdomains strength, hand function, mobility and activities of daily living after three months.

## Methods

### Study design

The prospective single-centre observational cohort study "Stroke-Cohort Augsburg (SCHANA Study)" is a collaboration project of the Chair of Epidemiology, University of Augsburg and the Department of Neurology and Clinical Neurophysiology at the University Hospital of Augsburg. A baseline interview was performed during the acute stroke hospital stay at the University Hospital Augsburg. Then a postal follow-up survey was conducted three months after hospital discharge. A further follow-up survey is being conducted 12 months after discharge.

The study started in September 2018 and patients were included until November 2019. Detailed information about the SCHANA study can be found in the publication of the study protocol [17].

Sample size was estimated based on the primary objectives of the SCHANA study, namely to investigate the impact of stroke treatment on recurrent events and stroke-related long-term survival [17]. A cumulative risk of stroke recurrence of 11% within one year was expected. With an estimated hazard ratio (HR) of 1.7 for the covariate of interest, a variance of 0.36 and a $rho^2 = 0.3$, at least 997 patients have to be included in the study to find significant differences with a statistical power of 80% (alpha = 5%).

Ethical approval was obtained from the ethics committee of the Ludwig-Maximilians-Universität München (No. 18–196) in May 2018. The data analysis of the present paper is restricted to baseline and three-month follow-up data available so far.

## Study population

Patients admitted to the University Hospital of Augsburg aged 18 years or older with a confirmed diagnosis of ischemic or haemorrhagic stroke were included in the study. Patients were excluded if they were not able to understand the consent form and answer the questions because of language difficulties and had no relatives available for translating.

Written informed consent was obtained from all participants or legal caregivers. If the patient was not able to give a self-report, a proxy interview was conducted.

## Survey data

The baseline questionnaire covered information about socio-demographics, social network, physical activity, depressiveness, general health status and smoking behaviour as detailed below. The follow-up questionnaire contained amongst other questions on the stroke-related quality of life.

**Physical activity.** The level of physical activity before the stroke event has been determined following the short form of the International Physical Activity Questionnaire (IPAQ) in the baseline survey [18, 19]. Information about the amount of time spent for walking or doing exhausting or moderate physical activity for at least ten minutes without interruption, was determined. The IPAQ score was categorized into low, moderate and high physical activity. Physical activity was classified as moderate, if vigorous-intensity activity of 20 minutes or more per day on at least three days per week or moderate-intensity activity on at least five days per week or at least 30 minutes walking per day or 600 or more MET (metabolic equivalent)-minutes per week were achieved by physical activity on at least five days. Physical activity was classified as high when at least 1500 MET-minutes per week were achieved by vigorous-intensity activity on at least three days or at least 3000 MET-minutes per week were achieved by physical activity on seven days. A low activity meant that none of the aforementioned criteria were met. More detailed information regarding the cut-off-points are given in the IPAQ manual [20].

**Socio-demographics and social network.** Date of birth, age, sex (male, female) and living situation were requested from the participants. The living situation was used as approximation for the social network of the participant. A variable that differs between solitarily and cohabiting living participants was created.

**Depressiveness.** The Patient Health Questionnaire (PHQ-9) [21–24] was applied to measure depressiveness. The scale values range from zero to 27. A value less than five can be interpreted as the absence of depressiveness. Values between five and ten constitute a mild degree of depressiveness. Values of ten and higher can be subdivided into moderate (ten to 14),

moderately severe (15 to 19), and severe (20 to 27) extent of depressiveness [25]. Meta-analyses of diagnostic validity studies showed that the PHQ-9 is a valid screening instrument in acute stroke patients and stroke survivors [26–28]. In contrast to other instruments, the nine-item questionnaire can be applied as self-report and has a low respondent burden due to its brevity. Thus, the PHQ-9 was considered as appropriate for assessing depressiveness in the in-hospital setting of the present study.

**General health status.**   Patients were asked to rate their general health status on a five-point Likert-scale ranging from "excellent" to "bad".

**Smoking behaviour.**   The questions on smoking history and status were adopted from the German National Cohort [29] and categorized into the three groups never, former and current smoker.

**Physical quality of life.**   Physical quality of life was assessed using the German version of the Stroke Impact Scale (SIS) 3.0 [30–34], which has been validated as a good measurement tool for health-related quality of life after stroke [35]. The SIS is a self-report questionnaire that evaluates disability and stroke-related quality of life after stroke [36]. It is sub-divided into the following eight dimensions of subjective health: strength, memory, emotions, communication, activities of daily living, mobility, hand function and participation [36]. Each subject includes several individual questions. Overall it contains 64 Likert-scaled questions, each with five points in terms of the difficulty experienced in completing the respective item in the past week [36]. Summative scores can be generated for each domain. The scores range from zero to 100 [36]. Higher values represent better health-related quality of life in the particular domain [37]. The four domains strength, hand function, mobility and activities of daily living can be combined to create a physical dimension score [37, 38]. The aggregation of the domain scores was done with an algorithm equivalent to the scoring algorithm of the SF-36 [37]. The domain score was defined as missing if at least half of the questions had missing responses [37]. The maximum score is 100 as well. The lowest score indicates severe restrictions in physical functioning, whereas the highest score indicates no restrictions [37]. In the present study, the physical domain scores strength, hand function, mobility and activities of daily living and the combined physical dimension score were used.

## Clinical data

Routinely collected clinical data were used to gather information on former stroke events, stroke severity, multimorbidity and body mass index (BMI).

**Stroke severity.**   To assess stroke severity, the National Institutes of Health Stroke Scale (NIHSS) and modified Rankin Scale (mRS) were used. The NIHSS assesses stroke severity by means of clinical symptoms [39]. It contains neurologic screenings and gathers information about the level of consciousness, gaze, visual fields, facial palsy, strength of the extremities, ataxia, sensory, language, dysarthria and extinction/inattention. Overall, it includes 15 items. The higher the total value, the greater is the degree of neurologic constraints [39].

The mRS assesses stroke severity by the degree of functional disability. The degree of disability ranges from zero (no symptoms) to six (dead) [39]. Although it is common to dichotomise the mRS levels [40], in this paper the ordinally form was used as this relates better to long-term outcomes and is therefore suggested to be preferred [41].

**Multimorbidity.**   With the information about pre-existing comorbidities, inference about multimorbidity was drawn. Considering the Charlson Comorbidity Index [42], that was already validated for studies with ischemic stroke patients [43], relevant diseases (e.g. hypertension, coronary heart disease, depression, diabetes mellitus) were summed up. If the number

of relevant diseases in addition to stroke was more than one, multimorbidity was ascertained [44]. A detailed list of the relevant diseases can be found in S1 Appendix.

## Data collection

All patients with stroke, who were admitted to the University Hospital of Augsburg from September 2018 to November 2019 were asked to participate in the SCHANA study. After informed consent a standardized computer-assisted baseline interview was performed by a trained study nurse.

A self-administered postal survey was used to collect follow up data three months after the patients discharge from the hospital. To minimize losses to follow-up, the participants were reminded by telephone in case of non-return of the questionnaire.

Data collection procedures were performed in accordance with the Declaration of Helsinki [45].

## Statistical analysis

Differences between the three subgroups of stroke patients in relation to their level of physical activity were investigated using analysis of variance. Depending on data structure and fulfilling of assumptions, Pearson chi-square or Kruskal-Wallis-Test were used. For the outcome variables, the two-sided, nonparametric Dwass-Steel-Critchlow-Fligner comparison procedure was used as post-hoc-test for identifying significant differences.

To determine the association between pre-stroke-physical activity and stroke-related quality of life, multiple linear regression models were carried out if the assumptions regarding the residuals and predictors were fulfilled. For each used SIS domain, a single regression model was carried out. The interaction effects of NIHSS, mRS, age or sex and physical activity were tested.

A quantile regression model for the quantiles 0.1 to 0.9 was calculated to gain a deeper understanding of the associations and to figure out whether the associations are particularly strong for special groups. Furthermore, the quantile regression was a useful tool to gain stable results although the distribution of the residuals did not show a perfect normal distribution.

For all tests an alpha level of 0.05 was defined and 95% confidence intervals were provided.

Data management and analysis was performed using SAS$^{®}$ Studio (Version 9.4).

**Selection of covariates.** A directed acyclic graph (DAG) (S1 Fig) was created to identify causal and non-causal structures, confounders and other types of bias as well as minimally sufficient adjustment sets [46, 47]. Literature-based expert knowledge and relevant findings from previous studies were used for creating the DAG (S1 Table). Accordingly, sex, age, former events, smoking, weight status, multimorbidity, depressiveness, general health status, social network and stroke severity were included as covariates to the DAG. This model represents the direct effect of pre-stroke physical activity on physical stroke related quality of life three months after stroke. The DAG was built with the web application 'DAGitty', a free software, licensed under the GNU general public licence (GPL) version 2 [48].

**Dealing with missing data.** From the entire dataset, observations with at least one missing source of information (baseline / follow-up questionnaire or medical chart), were excluded. Furthermore, observations were excluded if the exposure (IPAQ score) or all SIS domains of interest were missing (S2 Fig). In the multiple regression analysis, observations were excluded if one of the variables in the model contained a missing value.

**Validation of assumptions.** For metrically scaled variables without a normal distribution as well as for ordinally scaled variables, Kruskal-Wallis-Test was carried out. For nominally

scaled variables with more than five expected observations per cell Pearson chi-square test was calculated.

The six assumptions for the multiple regression models were tested. Quantile-quantile plots and partial regression models for metric variables were examined in order to check for a linear relationship between the dependent and the independent variables. Leverage Diagnosis Plot and the Cooks Distance were checked to ensure that there are no powerful outliers. Multicollinearity was tested by the variance inflation factor (VIF). Histograms and Scatterplots of the residuals were examined to check for homoscedasticity and normal distribution of the residuals. The Durbin-Watson-Test was calculated to check for independence of the residuals.

The quantile-quantile plots showed a positive but not perfect linear relationship for all dependent variables (SIS physical domain and the 4 subdomains) with the independent variables in the regression model. The variables EQ VAS and PHQ were squared and the variable age was squared and cubed, to improve the fulfilling of the model assumption of a linear relationship between the dependent and the independent variables.

## Results

The initial study sample comprised 858 patient datasets. Hereof 343 patients were excluded mainly because follow-up data was missing. For further details see S2 Fig.

Excluded participants (n = 361, 42.1%) had a mean age of 69.6 (median = 72.5) and 55.8% were male. Neither age nor sex differed significantly between included and excluded persons. Excluded participants had significantly higher NIHSS scores (mean = 3.8, median = 2), significantly higher mRS values (median = 3.0), a significantly higher general health status (mean = 3.4, median = 3.0) and significantly higher PHQ-scores (mean = 5.7, median = 5.0). The proportion of solitarily living participants was significantly higher in excluded participants (36.3%) and there were proportionally more participants with low physical activity (70.9%). Weight status, smoking status as well as the presence of multimorbidity and former stroke events did not significantly differ between the two groups. Detailed information regarding the analysis of differences of included and excluded patients is shown in S2 Table.

### Sample characteristics

The baseline characteristics of the analysis sample for the total group as well as stratified for the levels of physical activity are shown in Table 1. The cohort of 497 participants was mostly male (58.8%). The mean age was 69.6 ± 12.5 years. About 26% were living solitarily. A majority (78.5%) was multimorbid and about a fourth has sustained one or more former strokes. The median mRS score was 2 and the NIHSS averaged 2.8 (median = 1) in the total sample. Obesity (BMI $> = 30$ kg/m$^2$) was present in 25.9% of the participants and 14.8% stated to be current smokers. There were no significant differences between the three groups of physical activity for any of the examined variables, other than age, general health status and multimorbidity.

Table 2 shows the SIS scores raised at the three months follow-up. They are presented for the overall analysis sample as well as stratified by pre-stroke physical activity.

For all five SIS domains, significant differences between the three groups of physical activity were found. Subsequent Dwass-Steel-Critchlow-Fligner post-hoc-tests indicated that the differences were attributable to the groups moderate vs. low and low vs. high but not for moderate vs. high for all five domains. Detailed results of the post-hoc-tests are shown in S3 Table.

Table 2 demonstrates that the means of the SIS scores were very high and close to the maximum score of 100. The histograms of the SIS scores are presented in S3 Fig and show, that all scores were strongly skewed to the left.

**Table 1. Sample characteristics at baseline, overall and stratified by pre-stroke physical activity.**

| | | | Physical activity | | | | | |
| --- | --- | --- | --- | --- | --- | --- | --- | --- |
| | n | Total | Low | Moderate | High | test statistic | DF[2] | p-value |
| Variable | | 497 (100)[1] | 273 (54.9) | 94 (18.9) | 130 (26.2) | | | |
| Age in years, *mean (SD[3])* | 481 | 69.6 (12.5) | 71.6 (11.9) | 70.0 (12.3) | 65.4 (12.9) | 19.3 | 2 | < .0001[a] |
| Depressiveness: PHQ-Score, *mean (SD)* | 467 | 4.8 (4.3) | 5.0 (4.4) | 4.7 (4.4) | 4.6 (4.0) | 0.6 | 2 | 0.7588[a] |
| General health status, *mean (SD)* | 480 | 3.1 (0.9) | 3.2 (1.0) | 3.0 (0.9) | 2.8 (0.9) | 15.3 | 2 | 0.0005[a] |
| Stroke Severity | | | | | | | | |
| mRS[4], *median (IR[5])* | 488 | 2 (3.0) | 2 (3.0) | 2 (2.0) | 2 (2.0) | 3.8 | 2 | 0.1488[a] |
| NIHSS[6], *mean (SD)* | 487 | 2.8 (3.9) | 2.9 (3.6) | 2.8 (4.4) | 2.7 (4.3) | 3.8 | 2 | 0.1477[a] |
| Sex | 481 | | | | | | | |
| Male | | 283 (58.8) | 149 (58.0) | 61 (64.9) | 73 (56.2) | 1.9 | 2 | 0.3889[b] |
| Female | | 198 (41.2) | 108 (42.0) | 33 (35.1) | 57 (43.9) | | | |
| Social network | 472 | | | | | | | |
| Solitarily | | 123 (26.1) | 63 (25.3) | 23 (24.5) | 37 (28.7) | 0.7 | 2 | 0.7195[b] |
| Cohabiting | | 349 (73.9) | 186 (74.7) | 71 (75.5) | 92 (71.3) | | | |
| Weight status | 490 | | | | | | | |
| BMI[7] < 30 kg/m² | | 363 (74.1) | 196 (73.4) | 70 (74.5) | 97 (75.2) | 0.2 | 2 | 0.9261[b] |
| BMI ≥ 30 kg/m² | | 127 (25.9) | 71 (26.6) | 24 (25.5) | 32 (24.8) | | | |
| Smoking | 479 | | | | | | | |
| Current | | 71 (14.8) | 40 (15.7) | 13 (13.8) | 18 (13.9) | 1.7 | 4 | 0.7937[b] |
| Former | | 212 (44.3) | 112 (43.9) | 38 (40.4) | 62 (47.7) | | | |
| Never | | 196 (40.9) | 103 (40.4) | 43 (45.7) | 50 (38.5) | | | |
| Former stroke | 494 | | | | | | | |
| Yes | | 124 (25.1) | 72 (26.7) | 24 (25.5) | 28 (21.5) | 1.2 | 2 | 0.5382[b] |
| No | | 370 (74.9) | 198 (73.3) | 70 (74.5) | 102 (78.5) | | | |
| Multimorbidity | 497 | | | | | | | |
| Yes | | 390 (78.5) | 223 (81.7) | 75 (79.8) | 92 (70.8) | 6.3 | 2 | 0.0422[b] |
| No | | 107 (21.5) | 50 (18.3) | 19 (20.2) | 38 (29.2) | | | |

[1] Values are expressed as numbers (percentage) unless otherwise indicated.

[2] Degrees of Freedom

[3] Standard deviation

[4] modified Rankin Scale

[5] Interquartile range

[6] National Institutes of Health Stroke Scale

[7] Body Mass Index, BMI = kg/m²

[a] Kruskal-Wallis-Test

[b] Pearson chi-square

## Multiple linear regression analysis

To investigate the associations between pre-stroke physical activity and physical quality of life after three months, five linear regression analyses were performed: One for the SIS physical domain and one for each physical SIS subdomain. Durbin-Watson D depicts no autocorrelation for all five models. The interaction effects of NIHSS, mRS, age or sex and physical activity were not significant.

**SIS physical domain.** The results of the multiple linear regression model for the SIS physical domain are shown in Table 3. Multiple linear regression analysis was conducted with 437

**Table 2. Stroke Impact Scale scores at follow up, overall and stratified by pre-stroke physical activity.**

|  | n | Total | Physical activity | | | test statistic | DF[2] | p-value[a] |
|---|---|---|---|---|---|---|---|---|
|  |  |  | Low | Moderate | High |  |  |  |
| Stroke Impact Scale |  | 497 (100)[1] | 273 (54.9) | 94 (18.9) | 130 (26.2) |  |  |  |
| Physical domain, *mean (SD[3])* | 490 | 84.4 (19.0) | 80.6 (21.5) | 88.4 (13.6) | 89.5 (14.5) | 21.6 | 2 | <.0001 |
| Strength, *mean (SD)* | 401 | 72.8 (21.9) | 69.3 (22.1) | 76.7 (24.0) | 77.7 (18.6) | 14.6 | 2 | 0.0007 |
| Hand function, *mean (SD)* | 426 | 82.7 (25.4) | 78.3 (28.3) | 87.1 (20.7) | 88.9 (19.3) | 14.0 | 2 | 0.0009 |
| Mobility, *mean (SD)* | 489 | 85.2 (20.6) | 81.2 (23.9) | 89.7 (13.0) | 90.4 (15.1) | 14.6 | 2 | 0.0007 |
| Activities of daily living, *mean (SD)* | 493 | 86.8 (19.4) | 83.3 (22.1) | 90.2 (13.7) | 91.7 (15.2) | 17.9 | 2 | 0.0001 |

[1] Values are expressed as numbers (percentage) unless otherwise indicated.

[2] Degrees of Freedom

[3] Standard deviation

[a] Kruskal-Wallis-Test

[b] Pearson chi-square

(87.9%) patients with complete information on covariates. The table shows the coefficients for the SIS physical domain as well as for the included covariates.

The model with the SIS physical domain score as the dependent variable explained about 35% of the variance ($R^2 = 0.38$, adjusted $R^2 = 0.35$) and was significant (p = < .0001). The variable physical activity adjusted for age, sex, multimorbidity, general health, depressiveness, weight status, social network, smoking, former stroke events and stroke severity (NIHSS and mRS) showed a significantly linear correlation with the stroke-related quality of life in respect to the SIS physical domain three months after the stroke event. Both, moderate and high physical activity were significant (moderate: p = 0.0149 high: p = 0.0115). Patients with high physical activity had a 4.1 points higher score on the SIS physical domain scale than low active persons. Furthermore, moderate physical activity was associated with 4.3 points better physical quality of life after three months.

**SIS physical subdomains.** Table 4 summarises the results of the multiple linear regression analyses for the four subdomains strength, hand function, mobility and activities of daily living in terms of physical activity. All subdomains except SIS strength were significant. The results of the entire models can be found in S4–S7 Tables.

*SIS strength*. Multiple linear regression analysis was conducted with 355 (71.4%) patients with complete information on covariates. The model with the SIS strength score as the dependent variable explained about 19% of the variance ($R^2 = 0.24$, adjusted $R^2 = 0.19$) and was significant (p = < .0001). There was no significant linear correlation for physical activity adjusted for age, sex, multimorbidity, general health, depressiveness, weight status, social network, smoking, former stroke events and stroke severity (NIHSS and mRS) with the stroke-related quality of life in respect to SIS strength three months after the stroke event (moderate: p = 0.0931, high: p = 0.0633).

*SIS hand function*. Multiple linear regression analysis was conducted with 376 (75.7%) patients with complete information on covariates. The model with the SIS hand function score as the dependent variable explained about 25% of the variance ($R^2 = 0.30$, adjusted $R^2 = 0.25$) and was significant (p = < .0001). The variable physical activity adjusted for age, sex, multimorbidity, general health, depressiveness, weight status, social network, smoking, former stroke events and stroke severity (NIHSS and mRS) showed a significant linear correlation with the stroke-related quality of life in respect to SIS hand function three months after the stroke event. High physical activity was significant (p = 0.0298) while moderate physical

**Table 3. Associations between pre-stroke physical activity and physical quality of life after three months: Results of the multiple linear regression analysis.**

| Variable | Beta | (95% CI[1]) | p-value |
|---|---|---|---|
| Intercept | 192.9 | (102.5 to 283.4) | <0.0001 |
| Physical activity_high | 4.1 | (0.9 to 7.3) | 0.0115 |
| Physical activity_moderate | 4.3 | (0.8 to 7.7) | 0.0149 |
| Physical activity_low | Ref.[2] | | |
| Age | -5.9 | (-10.3 to -1.4) | 0.0097 |
| Age*Age[3] | 0.1 | (0.0 to 0.2) | 0.0050 |
| Age*Age*Age[4] | 0.0 | (0.0 to 0.0) | 0.0024 |
| Sex_female | 1.6 | (-1.3 to 4.4) | 0.2791 |
| Sex_male | 0.0 | | |
| Multimorbidity_no | -1.1 | (-4.5 to 2.2) | 0.5111 |
| Multimorbidity_yes | Ref. | | |
| EQVAS[5] | 6.7 | (0.2 to 13.3) | 0.0449 |
| EQVAS*EQVAS[6] | -2.0 | (-3.0 to -0.9) | 0.0003 |
| PHQ[7] | 0.5 | (-0.4 to 1.3) | 0.2906 |
| PHQ*PHQ[8] | -0.1 | (-0.1 to 0) | 0.0179 |
| BMI[9] < 30 | 1.7 | (-1.4 to 4.7) | 0.2779 |
| BMI ≥ 30 | 0.0 | | |
| Social network_cohabiting | 1.2 | (-2.0 to 4.3) | 0.4581 |
| Social network_solitarily | Ref. | | |
| Smoking_current | 2.5 | (-1.8 to 6.8) | 0.2597 |
| Smoking_former | 1.3 | (-1.7 to 4.3) | 0.3845 |
| Smoking_never | 0.0 | | |
| Former stroke_no | 4.2 | (1.1 to 7.4) | 0.0087 |
| Former stroke_yes | Ref. | | |
| NIHSS[10] | -0.9 | (-1.4 to -0.3) | 0.0016 |
| mRS[11]_1 | 0.5 | (-4.0 to 5.0) | 0.8312 |
| mRS_2 | -1.7 | (-5.9 to 2.4) | 0.4141 |
| mRS_3 | -6.4 | (-11.1 to -1.8) | 0.0070 |
| mRS_4 | -7.6 | (-12.8 to -2.4) | 0.0046 |
| mRS_5 | 3.5 | (-8.5 to 15.5) | 0.5654 |
| mRS_0 | Ref. | | |

[1] Confidence Interval

[2] Reference Group

[3] Age variable, squared

[4] Age variable, cubed

[5] European Quality of Life visual analogue scale (general health status)

[6] EQVAS variable, squared

[7] Patient Health Questionnaire (depressiveness)

[8] PHQ variable, squared

[9] Body Mass Index, BMI = kg/m$^2$

[10] National Institutes of Health Stroke Scale

[11] modified Rankin Scale

activity was not (p = 0.0565). Patients with high physical activity had a 5.6 points higher score on the SIS subdomain hand function scale than low active persons.

*SIS mobility*. Multiple linear regression analysis was conducted with 436 (87.7%) patients with complete information on covariates. The model with the SIS mobility score as the

Table 4. Associations between pre-stroke physical activity and the subdomains of physical quality of life after three months: Results of the multiple linear regression analysis.

| Stroke Impact Scale | Physical activity | | | | | |
| --- | --- | --- | --- | --- | --- | --- |
| | Moderate | | | High | | |
| | Beta[1] | (95% CI) | p-value | Beta | (95% CI) | p-value |
| Strength | 5.1 | (-0.3 to 10.5) | 0.0633 | 4.2 | (-0.7 to 9.1) | 0.0931 |
| Hand function | 5.3 | (-0.2 to 10.9) | 0.0565 | 5.6 | (0.6 to 10.7) | 0.0298 |
| Mobility | 4.9 | (1.2 to 8.7) | 0.0104 | 4.1 | (0.7 to 7.6) | 0.0200 |
| Activities of daily living | 3.3 | (-0.3 to 6.8) | 0.0729 | 3.7 | (0.4 to 7.0) | 0.0284 |

Models adjusted for age, sex, multimorbidity, general health, depressiveness, weight status, social network, smoking, former stroke events, stroke severity (NIHSS, mRS).
[1] Reference category for all variables: Low physical activity.

dependent variable explained about 35% of the variance ($R^2 = 0.39$, adjusted $R^2 = 0.35$) and was significant (p = < .0001). The variable physical activity adjusted for age, sex, multimorbidity, general health, depressiveness, weight status, social network, smoking, former stroke events and stroke severity (NIHSS and mRS) showed a significantly linear correlation with the stroke-related quality of life in respect to SIS mobility three months after the stroke event. Both moderate and high physical activity, were significant (moderate: p = 0.0104, high: p = 0.0200). Patients with high physical activity had a 4.1 points higher score on the SIS subdomain mobility scale than low active persons. Furthermore, moderate physical activity was associated with 4.9 points better physical quality of life after three months.

*SIS activities of daily living.* Multiple linear regression analysis was conducted with 440 (88.5%) patients with complete information on covariates. The model with the SIS activities of daily living score as the dependent variable explained about 29% of the variance ($R^2 = 0.32$, adjusted $R^2 = 0.29$) and was significant (p = < .0001). The variable physical activity adjusted for age, sex, multimorbidity, general health, depressiveness, weight status, social network, smoking, former stroke events and stroke severity (NIHSS and mRS) showed a significantly linear correlation with the stroke-related quality of life in respect to the SIS activities of daily living three months after the stroke event. High physical activity was significant (p = 0.0284) while moderate physical activity was not (p = 0.0729). Patients with high physical activity had a 3.7 points higher score on the SIS subdomain activities of daily living scale than low active persons.

## Quantile regression analysis

Compared to the multiple linear regression model, not one single, but nine sectionally regression coefficients were calculated for the quantiles 0.1 to 0.9 with the quantile regression. Fig 1 illustrates the regression coefficients for the two levels of physical activity for the SIS physical domain per quantile.

The regression coefficients were declining from the smaller to the higher quantiles. For the two lowest quantiles, the coefficients of moderate physical activity were 10.1 and 4.7, while it was 0.6 for quantile 0.9. This means, for patients with moderate physical activity, the physical quality of life turned out to be significantly higher for the patients with the 20% lowest SIS scores. The score of physical quality of life was 4.7 to 10.1 points higher compared to low active patients. In contrast, physical activity seemed not to have a decisive association for patients with higher post-stroke quality of life. The regression coefficient of high physical activity was 5.0 for quantile 0.1, whereby the confidence interval ranged from -2.0 to 12.0. For quantile 0.9 the regression coefficient was -0.2.

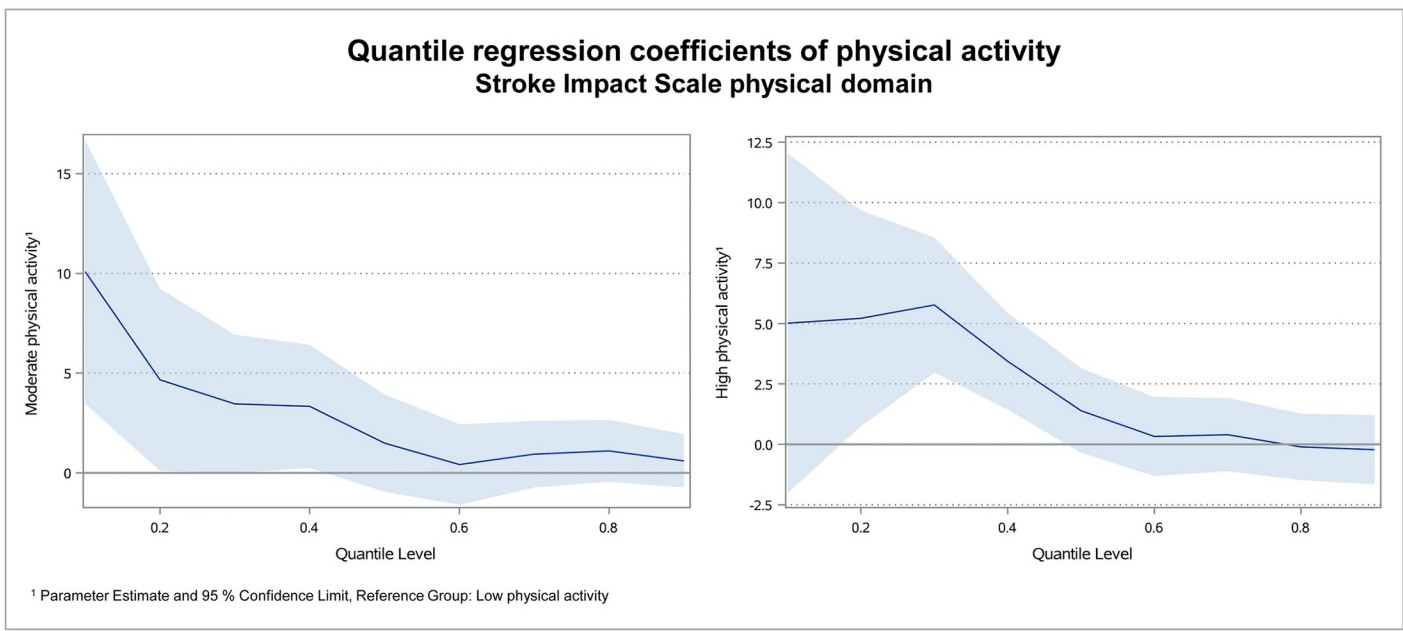

**Fig 1. Quantile regression coefficients of physical activity for the SIS physical domain.**

Fig 2 illustrates the regression coefficients of physical activity for SIS strength per quantile. In the subdomain strength, the coefficient of moderate physical activity was the lowest at quantile 0.1 (2.2). Afterwards it raised and reached its peak at quantile 0.3 (6.1). In the quantiles 0.4 to 0.9 it was varying between 3.2 and 5.1. In the group of high physical activity, the lowest coefficient was -1.9 in quantile 0.9.

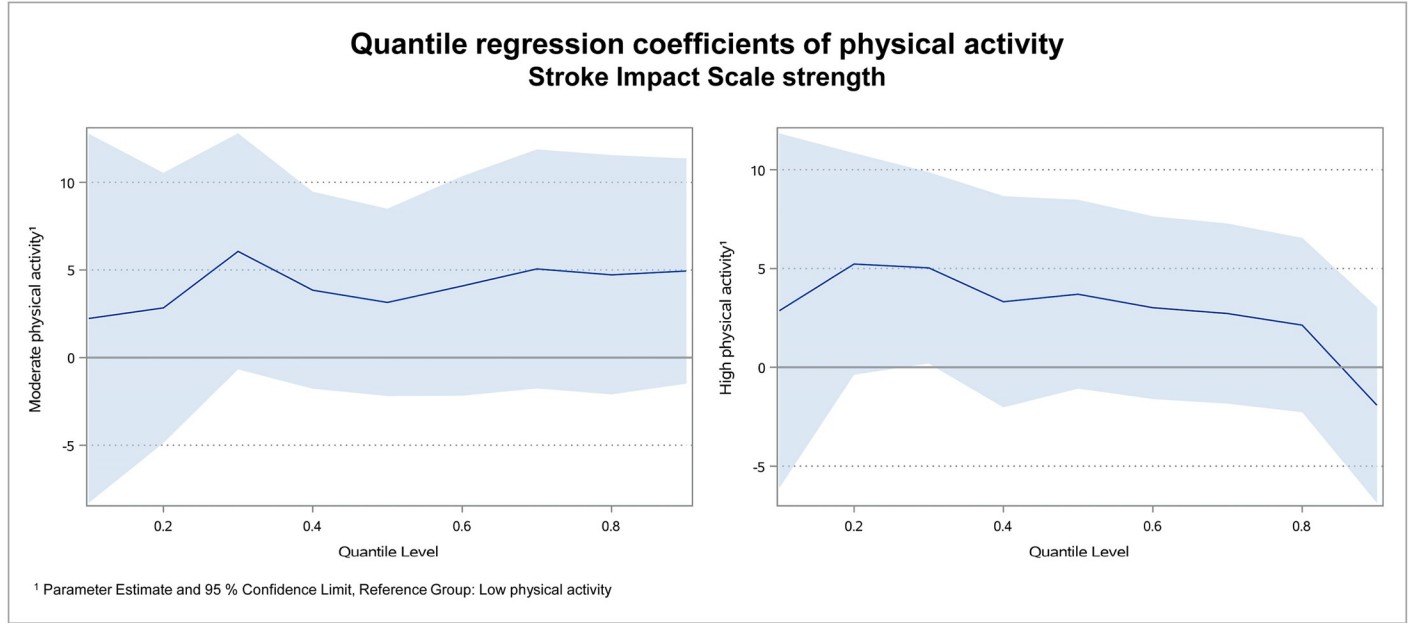

**Fig 2. Quantile regression coefficients of physical activity for SIS strength.**

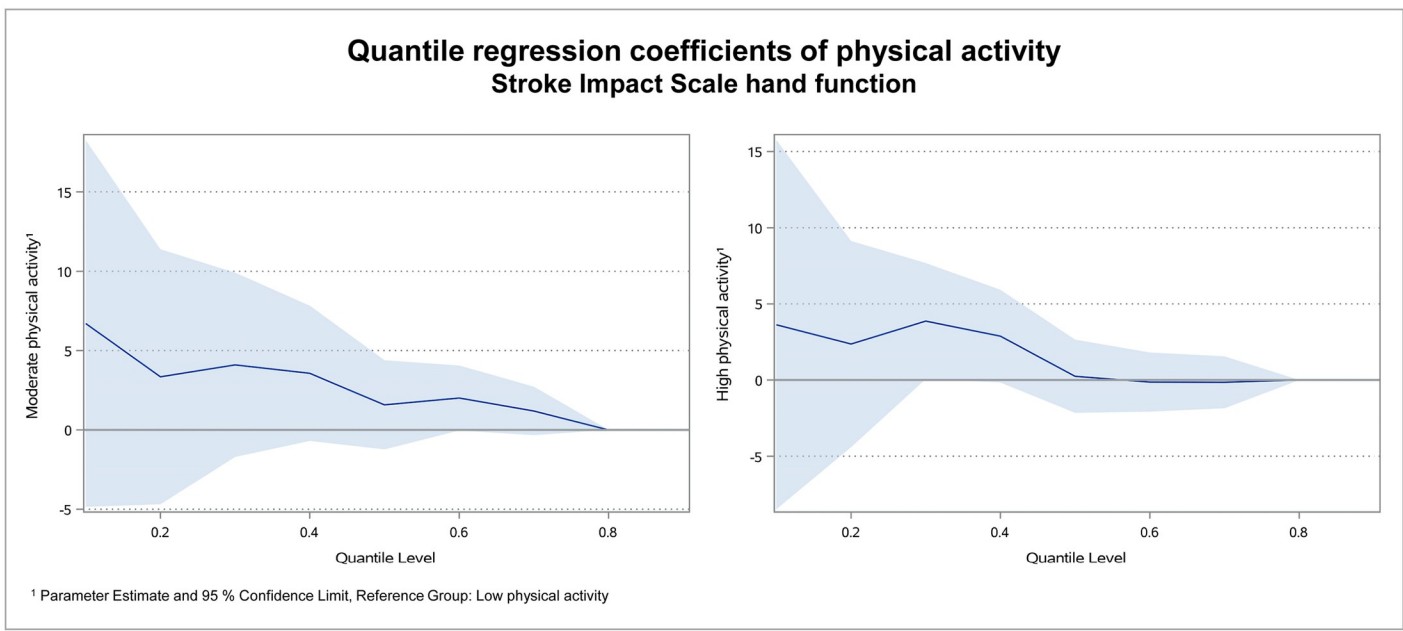

**Fig 3. Quantile regression coefficients of physical activity for SIS hand function.** In the subdomain hand function, the coefficient of moderate was 6.7 in quantile 0.1 and 1.2 in quantile 0.7. For high physical activity the coefficient was 3.6 in quantile 0.1 and -0.2 in quantile 0.7. Due to the high number of SIS hand function scores of 100, the maximum score of 100 was reached in quantile 0.7 and the quantile regression coefficients could not be calculated for quantiles 0.8 and 0.9.

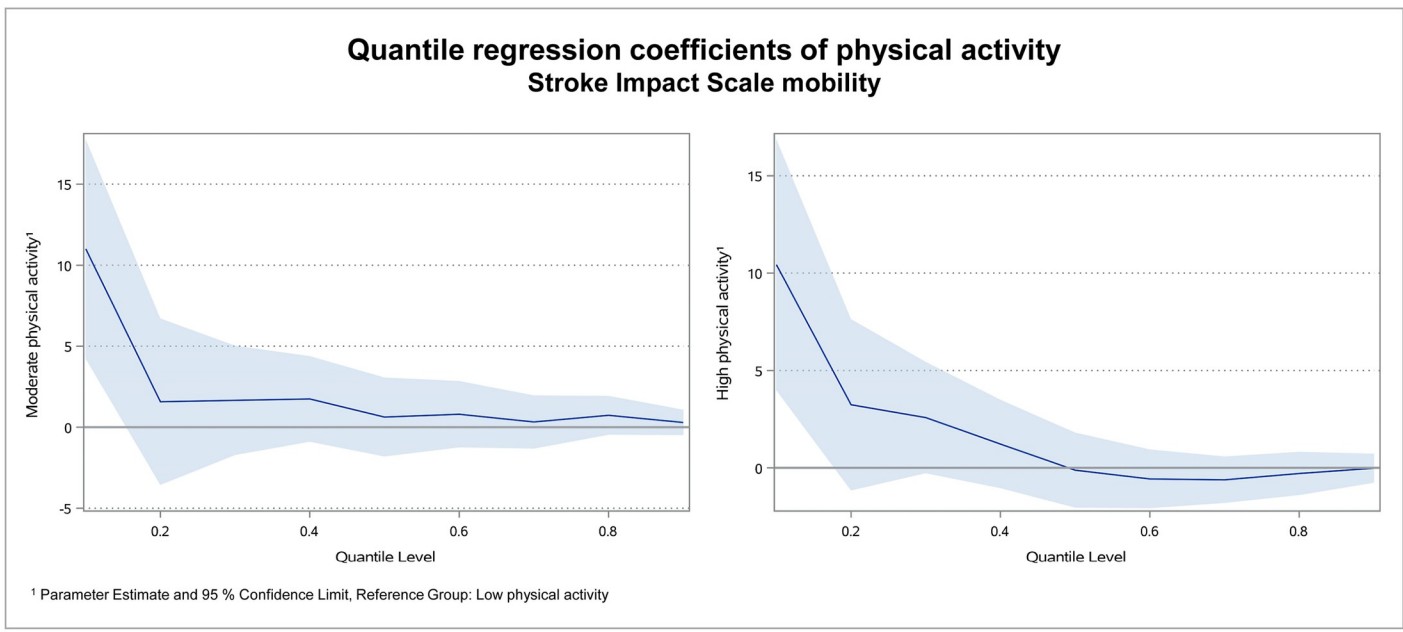

**Fig 4. Quantile regression coefficients of physical activity for SIS mobility.** In the subdomain mobility, the coefficient of moderate physical activity was 11.0 in quantile 0.1, while it declined to 1.7 or smaller in all other quantiles. For high physical activity it was 10.4 in quantile 0.1 and declined to 3.2 or smaller in all other quantiles. In the quantiles 0.5 to 0.9 the coefficient was slightly negative. This means, SIS scores turned out to be 11.0 points higher for moderate and 10.4 points higher for patients with high physical activity in the lowest quantile.

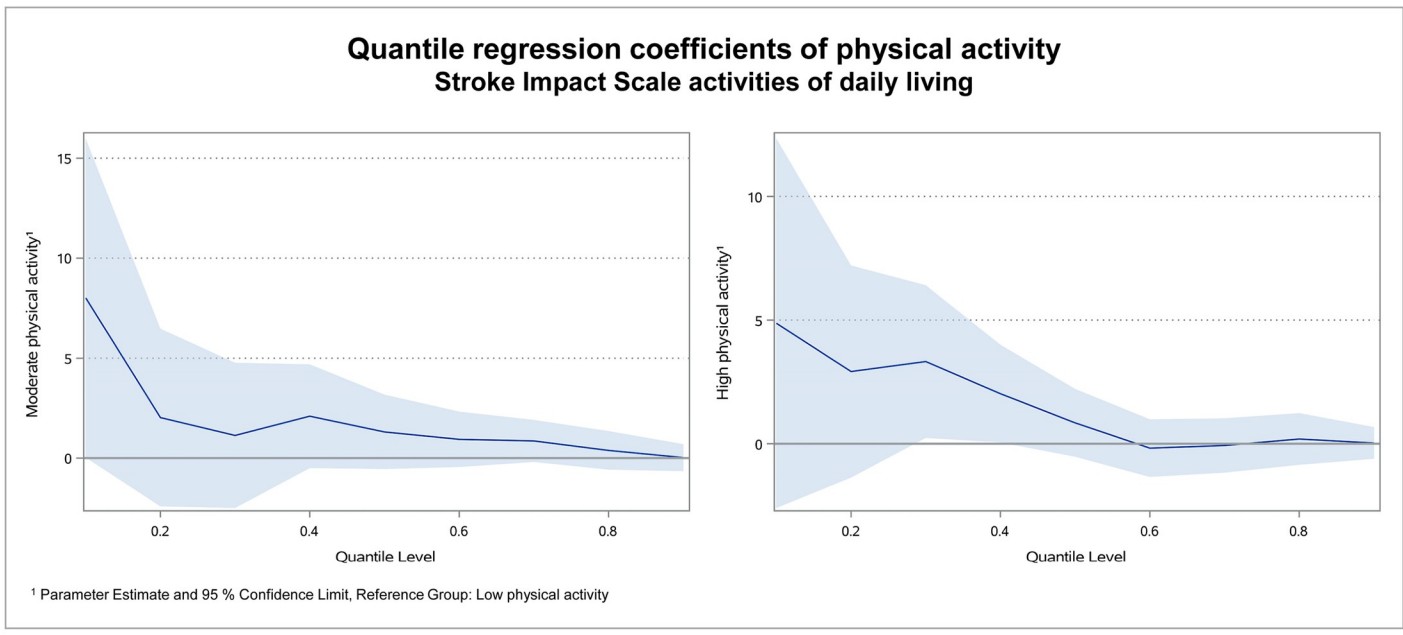

**Fig 5. Quantile regression coefficients of physical activity for SIS activities of daily living.** In the subdomain activities of daily living, the coefficient of moderate physical activity was 8.0 in quantile 0.1, while it was 2.1 or smaller in all other quantiles. For high physical activity the coefficient was 4.9 in quantile 0.1 and declined to 3.3 or smaller in all other quantiles. In the quantiles 0.6 and 0.7 it was slightly negative.

For the subdomains hand function, mobility and activities of daily living, the results were similar as for the physical domain score: Coefficients were high in the lower quantiles and declined to roughly zero in the higher quantiles. Figs 3–5 illustrate the regression coefficients of physical activity for SIS hand function, mobility and activities of daily living per quantile.

More detailed results of the quantile regression for all SIS domains can be found in S8 Table.

## Discussion

### Principal findings

The present study provided evidence for a significant association between pre-stroke physical activity and physical quality of life after three months in patients with mild disability. High pre-stroke physical activity was associated with a significantly better physical quality of life in the SIS physical domain as well as in the SIS subdomains hand function, mobility and activities of daily living. The results of the quantile regression analysis showed for all SIS domains, except for SIS strength, stronger effects of physical activity for patients in the lower quantiles compared to those in the higher quantiles of quality of life. For patients in the lower quantiles, who ranked their physical quality of life as bad, pre-stroke high or moderate physical activity was eminently favourable for their post-stroke physical quality of life. A possible explanation for this finding is that stronger trained muscles might be able to better compensate for the negative impact due to the stroke event. In addition, patients who were physically active before the stroke and perceive severe physical limitations may be more motivated to improve their impairments and to comply with physical therapy interventions in the rehabilitation phase after stroke. However, actual underlying reasons stay unclear and should be subject of further research. Furthermore, it must be mentioned, that the sample size in the lower quantiles was small, resulting in wide confidence intervals.

## Comparison with other studies and implications for research

Comparing the results of this study with findings from former studies is difficult due to the lack of research with similar issues. Studies which investigated physical activity in stroke patients had considerable variations in terms of study population and design, assessment of physical activity and outcomes [11–13, 49–51]. None of the former studies did explicitly investigate the physical post-stroke quality of life. However, the results of the studies using mRS as outcome could be compared with the results of the present study in an at least limited extent. It is obvious that the functional status and physical quality of life correlate with each other. For example, the need of assistance for performing body care (mRS score = 4) could result from restrictions in the different SIS domains like mobility, hand function or activities of daily living. One study showed that pre-stroke physical activity was associated with better long-term outcome and assessed the outcome using the mRS [12]. So did also a previous register study of the nationwide stroke database in Taiwan that examined the associations between pre-stroke physical activity and post-stroke functional status measured by mRS scores [49]. This study, which analysed nearly 40,000 stroke cases in Taiwan between 2006 and 2009, suggests that pre-stroke active persons had significantly better functional status at three months post-stroke [49]. Although functional status, measured by mRS scores, and not stroke-related quality of life was raised, parallels with the finding that pre-stroke physical activity is associated with better post-stroke physical quality of life, can be drawn.

Possible explanations for the benefits of pre-stroke physical activity for post-stroke physical quality of life could be greater knowledge about the positive health effects of physical activity and positive experiences with pre-stroke physical activity, which may support maintainance of higher levels of physical activity after stroke and as a consequence improve physical quality of life [52, 53]. The results of this study indicate that persons with low levels of pre-stroke physical activity are at risk of an impaired physical quality of life and may be supported by post-stroke counseling on the benefits of physical activity. Interventions may be offered in the setting of post-stroke rehabilitation.

However, further studies are needed to improve comparability of the results and confirm the findings of the present study by using same thresholds for physical activity. Also, future studies focussing on physical quality of life after stroke related to pre-stroke physical activity in patients with moderate and severe disability are needed.

## Strengths and limitations

To our knowledge, the present study is the first, which examined pre-stroke physical activity and post-stroke quality of life. The strengths of this study include the use of a longitudinal study design The use of patient self-reports assessed via a standardized personal interview and data from medical record provided elaborate data and enables comprehensive analysis. The covariates for the linear regression model were selected by methodologically sound techniques.

Potential limitations of this study should be considered in interpreting the results. A possible selection bias may have distorted the strength of the association between independent variables and outcome. Firstly, a number of hospitalized patients with stroke declined their participation, which may have led to an underrepresentation of severely affected patients in the study sample. Furthermore, several participants were excluded from the analyses due to missing follow-ups or data for the required variables. Since the study sample mainly included patients with mild degrees of functional impairments and disability, the results are only valid for this patient group and further studies on more severely affected patients are needed.

A second possible source of bias is that data from proxy interviews could have led to an overestimation of the association between pre-stroke physical activity and post-stroke quality of life in this study. A systematic review showed that proxy data for stroke severity often disagreed with data from patient self-reports and overestimated impairments compared with patient self-reports [33]. However, single studies showed that the usage of proxy reports were appropriate for research purposes to measure disability levels in stroke patients [54] and had an acceptable agreement for most SIS domains [30]. Significant differences between proxy data and SIS domains were considered as small and not clinically meaningful [55]. Furthermore, proxy data for physical activity appeared to be valid for persons with cognitive impairments [56].

Thirdly, the protective effect of physical activity may have been underestimated in the present analysis due to the tendency of over reporting physical activity as a desirable social behaviour [57] or due to a recall bias when reporting physical activity.

Finally, there are two possible confounders, which were not assessed in this study, namely post-stroke physical function and physical activity Systematic reviews, however, showed associations between post-stroke physical activity and physical function as well as quality of life [15, 58]. Thus, results of the present study should be interpreted with caution.

## Conclusion

In conclusion, pre-stroke physical activity seems to have an important and positive association with physical quality of life three months after stroke in patients with mild disability Particularly high effects were found for persons in the lower quantiles of physical quality of life. This epidemiological study disclosed the public health relevance of physical activity in this context, which has been hardly considered in research so far. Further studies are needed to deliver comparable results and to gain more comprehensive knowledge about the associations between pre-stroke physical activity and post-stroke physical quality of life.

## Supporting information

**S1 Fig. Directed acyclic graphs (DAG) of physical activity as independent variable and stroke related quality of life as outcome with covariates.**
(TIF)

**S2 Fig. Flow chart—Building the analysis sample.**
(TIF)

**S3 Fig. Histograms of the SIS scores.**
(TIF)

**S1 Table. References for directed acyclic graph (DAG).**
(DOCX)

**S2 Table. Differences of included and excluded patients.**
(DOCX)

**S3 Table. Results the post-hoc-tests: p-values for the comparison of the three groups of pre-stroke physical activity.**
(DOCX)

**S4 Table. Associations between pre-stroke physical activity and SIS strength after three months: Results of the multiple linear regression analysis.**
(DOCX)

**S5 Table. Associations between pre-stroke physical activity and SIS hand function after three months: Results of the multiple linear regression analysis.**
(DOCX)

**S6 Table. Associations between pre-stroke physical activity and SIS mobility after three months: Results of the multiple linear regression analysis.**
(DOCX)

**S7 Table. Associations between pre-stroke physical activity and SIS activities of daily living after three months: Results of the multiple linear regression analysis.**
(DOCX)

**S8 Table. Associations between pro-stroke physical activity and the four SIS subscales of physical quality of life after three months per quantile: Results of the quantile regression analysis.**
(DOCX)

**S1 Appendix. List of relevant diseases for defining multimorbidity.**
(PDF)

## Acknowledgments

The authors are grateful to all members of the department of Neurology and Clinical Neurophysiology at the University Hospital Augsburg, for their support. Moreover, we express our appreciation to all study participants.

## Author Contributions

**Conceptualization:** Melanie Zirnsak, Christine Meisinger, Jakob Linseisen, Michael Ertl, Philipp Zickler, Markus Naumann, Inge Kirchberger.

**Formal analysis:** Melanie Zirnsak.

**Investigation:** Christine Meisinger, Michael Ertl, Philipp Zickler.

**Methodology:** Christine Meisinger, Inge Kirchberger.

**Resources:** Jakob Linseisen, Markus Naumann.

**Supervision:** Christine Meisinger, Jakob Linseisen.

**Writing – original draft:** Melanie Zirnsak.

**Writing – review & editing:** Christine Meisinger, Jakob Linseisen, Michael Ertl, Philipp Zickler, Markus Naumann, Inge Kirchberger.

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
