## [Decision Letter · Decision Letter 0]

27 Aug 2021

PONE-D-21-16530

Associations between pre-stroke physical activity and physical quality of life three months after stroke

PLOS ONE

Dear Dr. Kirchberger,

Thank you for submitting your manuscript to PLOS ONE. After careful consideration, we feel that it has merit but does not fully meet PLOS ONE’s publication criteria as it currently stands. Therefore, we invite you to submit a revised version of the manuscript that addresses the points raised during the review process.

Some minor suggestions were presented by the reviewers, mainly in terms of the methods and instruments selected for outcomes measurement, therefore we suggest to read carefully each recommendation.

We look forward to receiving your revised manuscript.

Kind regards,

Miguel A. Barboza, MD, MSc

Academic Editor

PLOS ONE

1. Please ensure that your manuscript meets PLOS ONE's style requirements, including those for file naming. The PLOS ONE style templates can be found at https://journals.plos.org/plosone/s/file?id=wjVg/PLOSOne_formatting_sample_main_body.pdf and https://journals.plos.org/plosone/s/file?id=ba62/PLOSOne_formatting_sample_title_authors_affiliations.pdf.

Additional Editor Comments (if provided):

Reviewers' comments:

Reviewer's Responses to Questions

**Comments to the Author**

1. Is the manuscript technically sound, and do the data support the conclusions?

Reviewer #1: Partly

Reviewer #2: Yes

Reviewer #3: Yes

2. Has the statistical analysis been performed appropriately and rigorously? 

Reviewer #1: No

Reviewer #2: Yes

Reviewer #3: Yes

3. Have the authors made all data underlying the findings in their manuscript fully available?

Reviewer #1: Yes

Reviewer #2: Yes

Reviewer #3: Yes

4. Is the manuscript presented in an intelligible fashion and written in standard English?

Reviewer #1: Yes

Reviewer #2: Yes

Reviewer #3: Yes

5. Review Comments to the Author

Reviewer #1: Major comments:

1. Complete case analysis was used to handle missing data. Unfortunately there are significant differences in stroke severity and post stroke disability between the complete cases and missing data cases. Though this has been reported as a limitation of the study the authors cannot report that there is 'compelling' evidence of association between pre-stroke activity levels and post-stroke QOL. Perhaps authors can reconsider reframing this paper as "Associations between pre-stroke physical activity and physical quality of life three months after stroke amongst stroke survivors with mild disability" and rewriting their discussion and conclusion to reflect the biases in the study.

2. There is much literature demonstrating the significant association between post-stroke physical function and QOL. Physical function is an important confounder that should have been adjusted for. This could also bias the associations found.

3. The authors also need to present some existing evidence or hypothesis as to why pre-stroke physical activity has associations with post stroke QOL? and how will the findings be useful and why is this an important question?

Reviewer #2: How was the sample size calculated?

On page 2, line 29, after "Germany", there should be a ",", instead of ".".

In line 41, "the effect was particularly high for persons in the lower quantiles of physical quality"; I suggest changing the wording as it is not entirely clear what it means.

The work is novel and interesting, with details about the analysis, an exhaustive review of the limitations, and with conclusions according to the results.

Reviewer #3: The Patient Health Questionnaire (PHQ-9) is a widely used screening tool for major depressive disorder, although there is debate surrounding its diagnostic properties, ¿why did you choose that scale and how do you answer it in patients with aphasia?.

In Stroke Impact Scale ¿do you evaluated if there is any difference between the responses of caregivers and patients?

¿Do you considered if the patient had any previous cognitive impairment?

6. PLOS authors have the option to publish the peer review history of their article (what does this mean?). If published, this will include your full peer review and any attached files.

Reviewer #1: No

Reviewer #2: No

Reviewer #3: **Yes: **PAULINA DE REGIL GONZALEZ

---

## [Author Response · Author response to Decision Letter 0]

29 Sep 2021

Reviewer #1: 

1. Complete case analysis was used to handle missing data. Unfortunately there are significant differences in stroke severity and post stroke disability between the complete cases and missing data cases. Though this has been reported as a limitation of the study the authors cannot report that there is 'compelling' evidence of association between pre-stroke activity levels and post-stroke QOL. Perhaps authors can reconsider reframing this paper as "Associations between pre-stroke physical activity and physical quality of life three months after stroke amongst stroke survivors with mild disability" and rewriting their discussion and conclusion to reflect the biases in the study.

Thank you very much for this important comment. We appreciate your suggestions. Accordingly we have changed the title of the paper to better reflect the study sample characteristics: „Associations between pre-stroke physical activity and physical quality of life three months after stroke in patients with mild disability.“

We have also modified the discussion: 

„The present study provided evidence for a significant association between pre-stroke physical activity and physical quality of life after three months in patients with mild disability.“

„Also, future studies focussing on physical quality of life after stroke related to pre-stroke physical activity in patients with moderate and severe disability are needed.“

 „Since the study sample mainly included patients with mild degrees of functional impairments and disability, the results are only valid for this patient group and further studies on more severely affected patients are needed.“

Finally, we have modified the conclusions:

„In conclusion, pre-stroke physical activity seems to have an important and positive association with physical quality of life three months after stroke in patients with mild disability.”

2. There is much literature demonstrating the significant association between post-stroke physical function and QOL. Physical function is an important confounder that should have been adjusted for. This could also bias the associations found.

We agree with you, that the degree of physical impairment may be an important confounder, which could not be considered in this study.. We have included this limitation in the discussion section:

„Finally, there are two potential confounders which were not assessed in this study, namely post-stroke physical function and physical activity. Systematic reviews, however, showed associations between post-stroke physical activity and physical function as well as quality of life [3,4]. Thus, results of the present study should be interpreted with caution.“

3. The authors also need to present some existing evidence or hypothesis as to why pre-stroke physical activity has associations with post stroke QOL? and how will the findings be useful and why is this an important question?

We have modified the introduction in order to address your valuable comment:

“Regular physical activity decreases stroke incidence [6,7] and was associated with better cognitive function [8] and even fewer symptoms of depression [9] in those affected. There are a few studies which reported a significant association between pre-stroke physical activity and post-stroke functional status as assessed by the National Institutes of Health Stroke Scale (NIHSS), the Modified Ranking Scale (mRS) and the Barthel Index [10-13]. In addition, low level of physical activity before stroke predicted low physical activity after stroke [14]. This is important, since higher levels of post-stroke physical activity are related with better physical function as well as better quality of life [15]. However, little is known about the correlation between pre-stroke physical activity and post-stroke outcomes such as health-related quality of life (HRQOL), which is considered to be considerably lower in stroke survivors than population norm [16] . Since physical activity is an essential target of stroke rehabilitation, further knowledge about the relation of pre-stroke physical activity and post-stroke HRQOL could be used to identify patients at risk for inactivity and impaired HRQOL after stroke.”

Furthermore, we have addedd the following paragraph to the discussion section:

„Possible explanations for the benefits of pre-stroke physical activity for post-stroke physical quality of life could be greater knowledge about the positive health effects of physical activity and positive experiences with pre-stroke physical activity, which may support maintainance of higher levels of physical activity after stroke and as a consequence improve physical quality of life [52,53]. The results of this study indicate that persons with low levels of pre-stroke physical activity are at risk of an impaired physical quality of life and may be supported by post-stroke counseling on the benefits of physical activity. Interventions may be offered in the setting of post-stroke rehabilitation.“

Reviewer #2: 

How was the sample size calculated?

We included the following paragraph in the methods section: 

„Sample size was estimated based on the primary objectives of the SCHANA cohort study, namely to investigate the impact of stroke treatment on recurrent events and stroke-related long-term survival [17]. A cumulative risk of stroke recurrence of 11% within one year was expected. With an estimated hazard ratio (HR) of 1.7 for the covariate of interest, a variance of 0.36 and a rho2 = 0.3, at least 997 patients have to be included in the study to find significant differences with a statistical power of 80% (alpha = 5%).“

On page 2, line 29, after "Germany", there should be a ",", instead of ".".

We have corrected this typo. 

In line 41, "the effect was particularly high for persons in the lower quantiles of physical quality"; I suggest changing the wording as it is not entirely clear what it means.

We have modified this sentence accordingly. It now reads:

“… the effect was particularly strong for persons with low physical quality of life after three months.”

Reviewer #3: 

The Patient Health Questionnaire (PHQ-9) is a widely used screening tool for major depressive disorder, although there is debate surrounding its diagnostic properties, ¿why did you choose that scale and how do you answer it in patients with aphasia?.

The PHQ-9 was selected based on available metanalyses and systematic reviews which showed that the PHQ-9 is an approriate measure for screening depression in stroke patients Contrary to other available measures such as the The Center of Epidemiological Studies-Depression Scale (CESD) or the Hamilton Depression Rating Scale (HDRS), the PHQ-9 is a self-report questionnaire and is very short (9 items compared with 20 items (CESD) and 17 items (HDRS)) and therefore feasible for its use in the hospital setting. We have now added some information on the reasons for selecting the PHQ-9 in the methods section:

“In contrast to other instruments, the nine-item questionnaire can be applied as self-report and has a low respondent burden due to its brevity. Thus, the PHQ-9 was considered as appropriate for assessing depressiveness in the in-hospital setting of the present study.”

”

For patients with aphasia, caregivers could complete the questionnaires.

In Stroke Impact Scale ¿do you evaluated if there is any difference between the responses of caregivers and patients?

Thank you for this important comment. Unfortunately, in our study no information was obtained whether the questionnaires were filled in by the patients themselves or their caregivers. However, since the sample consisted mainly of patients with mild disabilities according to NIHSS and mRS, we assume that most patients were able to provide a self-report. In addition, studies on differences between self- and caregiver reports in stroke patients indicated acceptable comparability of scores derived from the Stroke Impact Scale (e.g. Duncan PW, Lai SM, Tyler D, Perera S, Reker DM, Studenski S. Evaluation of proxy responses to the Stroke Impact Scale. Stroke. 2002; 33:2593–9.)

¿Do you considered if the patient had any previous cognitive impairment? 

Previous cognitive impairment was not assessed.

---

## [Decision Letter · Decision Letter 1]

21 Mar 2022

Associations between pre-stroke physical activity and physical quality of life three months after stroke in patients with mild disability.

PONE-D-21-16530R1

Dear Dr. Kirchberger,

We’re pleased to inform you that your manuscript has been judged scientifically suitable for publication and will be formally accepted for publication once it meets all outstanding technical requirements.

Kind regards,

George Vousden

Deputy Editor-in-Chief

PLOS ONE

Additional Editor Comments (optional):

Reviewers' comments:

Reviewer's Responses to Questions

**Comments to the Author**

1. If the authors have adequately addressed your comments raised in a previous round of review and you feel that this manuscript is now acceptable for publication, you may indicate that here to bypass the “Comments to the Author” section, enter your conflict of interest statement in the “Confidential to Editor” section, and submit your "Accept" recommendation.

Reviewer #1: All comments have been addressed

Reviewer #2: All comments have been addressed

Reviewer #3: All comments have been addressed

2. Is the manuscript technically sound, and do the data support the conclusions?

Reviewer #1: Yes

Reviewer #2: Yes

Reviewer #3: Yes

3. Has the statistical analysis been performed appropriately and rigorously? 

Reviewer #1: Yes

Reviewer #2: Yes

Reviewer #3: Yes

4. Have the authors made all data underlying the findings in their manuscript fully available?

Reviewer #1: Yes

Reviewer #2: Yes

Reviewer #3: Yes

5. Is the manuscript presented in an intelligible fashion and written in standard English?

Reviewer #1: Yes

Reviewer #2: Yes

Reviewer #3: Yes

6. Review Comments to the Author

Reviewer #1: Pls ensure referencing is as per journal requirements. In the current submitted manuscript, the in text referencing is not a superscript.

Reviewer #2: The requested modifications have been adequately answered. The methodology is described, and the conclusions are based on the findings.

Reviewer #3: (No Response)

7. PLOS authors have the option to publish the peer review history of their article (what does this mean?). If published, this will include your full peer review and any attached files.

Reviewer #1: No

Reviewer #2: **Yes: **Pavel Loeza Magaña

Reviewer #3: **Yes: **PAULINA DE REGIL GONZALEZ

---

## [Editor Report · Acceptance letter]

28 Mar 2022

PONE-D-21-16530R1 

Associations between pre-stroke physical activity and physical quality of life three months after stroke in patients with mild disability. 

Dear Dr. Kirchberger:

I'm pleased to inform you that your manuscript has been deemed suitable for publication in PLOS ONE. Congratulations! Your manuscript is now with our production department. 

Kind regards, 

on behalf of

Dr. George Vousden 

Staff Editor

PLOS ONE